# Impairment in facial expression generation in patients with repaired unilateral cleft lip: Effects of the physical properties of facial soft tissues

Donghoon Lee[1], Chihiro Tanikawa[1,2]*, Takashi Yamashiro[1]

**1** Graduate School of Dentistry, Osaka University, Suita, Osaka, Japan, **2** Center for Advanced Medical Engineering and Informatics, Osaka University, Suita, Osaka Japan

* ctanika@dent.osaka-u.ac.jp, ctanika@gmail.com

**Data Availability Statement:** All data files are available from the DRYAD database (doi:10.5061/dryad.h44j0zpjp).

## Abstract

Patients with repaired unilateral cleft lip with palate (UCLP) often show dysmorphology and distorted facial motion clinically, which can cause psychological issues. However, no report has clarified the details concerning distorted facial motion and the corresponding possible causative factors. In this study, we hypothesized that the physical properties of the scar and surrounding facial soft tissue might affect facial displacement while smiling in patients with UCLP (Cleft group). We thus examined the three-dimensional (3D) facial displacement while smiling in the Cleft and Control groups in order to determine whether or not the physical properties of facial soft tissues differ between the Cleft and Control groups and to examine the relationship between the physical properties of facial soft tissues on 3D facial displacement while smiling. Three-dimensional images at rest and while smiling as well as the facial physical properties (e.g. viscoelasticity) of both groups were recorded. Differences in terms of physical properties and facial displacement while smiling between the two groups were examined. To examine the relationship between facial surface displacement while smiling and physical properties, a canonical correlation analysis (CCA) was conducted. As a result, three typical abnormal features of smiling in the Cleft group compared with the Control group were noted: less upward and backward displacement on the scar area, downward movement of the lower lip, and a greater asymmetric displacement, including greater lateral displacement of the subalar on the cleft side while smiling and greater alar backward displacement on the non-cleft side. The Cleft group also showed greater elastic modulus at the upper lip on the cleft side, suggesting hardened soft tissue at the scar. The CCA showed that this hard scar significantly affected facial displacement, inducing less upward and backward displacement on the scar area and downward movement of the lower lip in patients with UCLP (correlation coefficient = 0.82, p = 0.04); however, there was no significant relationship between greater nasal alar lateral movement and physical properties of the skin at the scar. Based on these results, personalizing treatment options for dysfunction in facial expression generation may require quantification of the 3D facial morphology and physical properties of facial soft tissues.

**Funding:** This work was partially supported by the Japan Society for the Promotion of Science (JSPS) KAKENHI (grant nos. 22792048 and 25862008), and the Center of Innovation Program from Japan Science and Technology Agency (JST). CT received these grants from JSPS and JST. We declare that the JSPS and JST had no role in the study design, data collection and analysis, decision to publish, or preparation of the manuscript. 【URLs】 JSPS https://www.jsps.go.jp/english/index.html JST https://www.jst.go.jp/EN/.

**Competing interests:** The authors have declared that no competing interests exist.

## Introduction

A cleft lip and palate (CLP) is the most common orofacial congenital disease and it is characterized by openings or splits in the upper lip, palate, or both when birth (1/700) [1]. Although patients with CLP undergo primary lip repair usually at two to four months of age, the affected patients experience a number of problems, including dental, speech, hearing, and aesthetic complications. Among these complications, issues with the facial appearance, including the movement of the face, are considered the most challenging to address, as lip surgeries leave a scar on the lip, and dysmorphic features remain even after surgery. In addition, facial expressions play an important role in nonverbal communication, conveying emotions and thoughts, and it is no exaggeration to say that facial expression is a social function. Facial expressions thus have a strong influence on an individual's perceived self-worth. From a sociopsychological perspective, achieving an acceptable facial appearance and facial movements in patients with CLP is a crucial treatment goal in surgical/orthodontic clinics.

To quantify these issues associated with facial dysmorphology and dysfunction, several studies have reported on distorted facial movements during facial expression (e.g. smiling) and functional impairments in patients with CLP [2–4]. A previous study [2] employed two-dimensional photographs of patients before and after smiling from the frontal view and reported a greater displacement on the cleft side in the lateral direction than in the control group. A three-dimensional (3D) study using a video-based tracking system showed that lateral movements of the upper lip were greater than vertical movements in patients with cleft lip with or without revisions [4]. In that study, the revision and nonrevision groups demonstrated 6% to 28% less upper lip movements, with smiles having the most restriction in movement and greater asymmetry of the upper lip movement than in the noncleft group.

However, the causative factors underlying these distorted facial motions in patients with UCLP have been unclear. Several potential causative factors have been proposed [5], including muscle impairment [4], scar tissue [6], and an abnormal neurosensory function [7]. Scar formation is the most likely candidate, as it results from collagen fiber synthesis and alignment in the presence of a tensile stress field generated by wound contraction from prior cleft repair [8], which may result in the abnormal movement of the face during smiling. Tightening of the cleft scar on the upper lip may pull the nose and surrounding upper lip area toward the affected side while smiling, thus causing asymmetric lateral displacement of the alar and upper lip areas. Indeed, a previous study that examined perioral stiffness using a face-referenced measurement technology known as OroSTIFF [6] showed that patients with a repaired cleft lip who did not have lip revision surgery had greater mean inter-angle stiffness scores than noncleft patients. However, the direct relationship between the physical properties of the perioral area and the distorted facial movement during facial expression remains unclear.

Therefore, in the present study, we hypothesized that the physical property of soft tissue around scarring in the nasolabial region would restrict the facial functional movement during facial expression in patients with CLP.

When measuring the physical properties of the nasolabial region, the mechanical behavior of the skin exhibits a time-dependent response, as a consequence of its viscoelastic characteristics [9–11]. Viscoelasticity is the property of materials that exhibit both viscous and elastic characteristics when undergoing deformation, demonstrating both creep and stress relaxation [12]. Several non-invasive and clinically simple instruments for measuring the viscoelastic properties of the soft tissue have recently been developed. For example, a previous study [9] measured the elasticity of the skin of the face and ventral forearm in 170 women with non-invasive suction-type instruments (Cutometer®) and evaluated the effects of age and exposure to sunlight. Another study measured cutaneous flexibility at anatomic areas on normal

volunteers and burn scars using an instrument (Pneumatonometer®) that had originally been used to measure the intraocular pressure [10]. More recently, a previous study [11] measured the physical properties of the skin in patients with systemic sclerosis using a non-invasive portable sensing device (Vesmeter®) and reported an increased elastic modulus of the affected skin with systemic sclerosis. Furthermore, previous studies have shown an increased elastic modulus of scar tissue of body skin and reported that an increased elastic modulus was related to less extensibility of the arm [12, 13]. Given the above, we suspected that measuring the viscoelasticity would clarify the effects of the scar conditions on the facial movement in a site-specific manner.

With regard to measuring the facial functional movement during facial expressions, 3D digital camera systems have recently become a useful clinical tool for the quantification of facial surfaces. For example, a 3D digital camera system was used for the evaluation of changes in the nasal morphology after maxillomandibular surgery [14], the evaluation of social smile reproducibility [15], the assessment of facial outcome following orthognathic treatment in patients with craniofacial syndrome [16], the recognition of differences between patients with craniofacial syndrome and healthy controls [17], and the assessment of volumetric changes due to revision surgery in the nose of patients with CLP [18]. Furthermore, a recent study demonstrated a better intermediate precision of maximum smiling on 3D facial surfaces [19]. In that study, the maximum-effort smile provided better test-retest reliability than a posed/social smile, and the maximum-effort smile can be applied clinically to estimate the capability of lip movement. Therefore, in the present study, we employed 3D facial surfaces at rest and at maximum smiling in order to comprehensively examine the restricted facial functional movement during facial expressions.

The present study (1) examined whether or not the 3D facial displacement while smiling in patients with a repaired UCLP differed from that of healthy adults, (2) investigated whether or not the physical properties of facial soft tissue differed between patients with a repaired UCLP and healthy adults, and (3) evaluated the relationship between the physical properties of facial soft tissues on 3D facial displacement while smiling in patients with repaired UCLP in a site-specific manner.

## Materials and methods

The study protocol was approved by the Ethics Committee of the Graduate School of Dentistry, Osaka University (IRB No. H20-E19-2). An institutional review board-approved written informed consent form was distributed to and signed by all participants prior to their involvement in the study. If the participant was under the age of 20, we obtained consent from parents or guardians of the minors included in the study.

### Participants

Japanese patients with a repaired UCLP (Cleft group; n = 41, mean age = 21.46 ± 4.27 years old, 21 men and 20 women) and healthy adults with a straight-type facial profile and normal occlusion (Control group; n = 41, mean age = 25.78 ± 3.35 years old, 21 men and 20 women) were enrolled in the present study.

The inclusion criteria of the Cleft group were as follows: age 15–37 years old, no facial paralysis, no history of any psychiatric disorder, no subjectively or objectively discernible jaw dysfunction, body mass index (BMI) of 18.50–24.99 [20], and no maxillofacial plastic surgeries in the past 6 months. The inclusion criteria of the Control group were as follows: age 18–35 years old, no congenital facial deformities (including a cleft lip or palate), no facial paralysis, no noticeable scars or skin diseases of the neck or dentofacial regions (or history thereof), no

history of any psychiatric disorder, no subjectively or objectively discernible jaw dysfunction, a BMI of 18.50–24.99 [20], an overbite of 1.0–5.0 mm, an overjet of 0.0–7.0 mm, and a straight-type facial profile.

## Data acquisition

The subjects were asked to sit on a fixed chair with a natural head position without head support. They were then asked to perform tasks as described in Table 1 (i.e. at rest and at the peak of the maximum smile) following our previous study [19], where the maximum smile provided substantial to almost perfect agreement with the repeated measures of the rest posture and maximum effort smile (intraclass correlation coefficient 0.60 to $\leq$1.00). The 95% confidence interval minimal detectable change for the repeated measures of the rest posture and the maximum effort smile exhibited means of 0.8 and 1.3 mm, respectively, on the z-axis. The subjects were instructed orally for each task and asked to maintain the expressions for approximately 2 s. After several rehearsals, each expression was recorded once with a 3D image capturing device (3-DMDcranial System; 3-DMD, Atlanta, GA, USA). Each type of expression was recorded with a resting interval of approximately 20 s between expressions. The experimenter (D. L.) controlled the system from a position out of the subject's view.

The elastic modulus ($kN/m^2$) and viscosity coefficient ($N{\cdot}s/m^2$) of facial landmarks, including the cheek (Chk), crista philtri superior' (Cphs'), crista philtri inferior (Cphi), and cheilion (Ch) on the left side, were also measured using a viscoelasticity measuring instrument (Ves-meter-E100Hs, WaveCyber Corp., Saitama, Japan, Fig 1 [11]). This device consists of a probe with a built-in position sensor and computer [11]. When the probe is placed at a right angle on the skin, the indenter is depressed against the skin at a constant speed by electromagnetic force, and the path of the indenter is constantly traced by the position sensor. The instrument measures the hardness of an object as the area of the depression divided by the pressure of the indenter, and calculates elasticity and viscosity by analyzing the waveform of the stress relaxation behaviors of the skin. That is, the instrument analyzes the stress relaxation behavior of viscoelastic materials using the Voigt model, which consists of two components, a purely viscous dashpot and a purely elastic spring connected in parallel. Prior to the measurements, we examined test-retest reliability of this instrument, as shown in S1 Text and S1 Table.

The definition of Cphi and Ch was based on the anthropometric investigations described by Mulliken et al. [21]. Cphs' was newly defined in the present study as the point on the philtral crest approximately 12 mm below the conventional Cphs. This is because the probe of the viscoelasticity-measuring instrument whose radius was 12 mm cannot reach the Cphs under the nose. Chk was defined as the most prominent point of the cheek. The room temperature was set at 25°C.

## Data processing of 3D images

A coordinate system of the 3D images recorded at rest was established based on our previous studies (S1 Fig) [22, 23]. The images recorded at the peak of the maximum smile were also

**Table 1. Definition of tasks.**

| Task | Definition |
|---|---|
| **Rest** | After swallowing saliva, subjects assumed a relaxed facial position with the lips in repose and the teeth in light contact in the habitual maximum intercuspation position. Recording was made approximately 10 s after commencement of saliva swallowing. |
| **Peak of the maximum smile** | A grinning effort was made with the corners of the mouth pulled laterally and the cheeks elevated with maximal effort and with the teeth in the habitual maximum intercuspation position. |

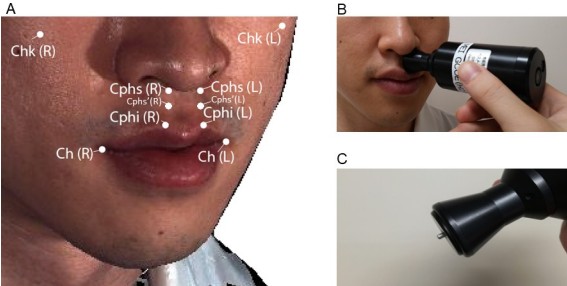

**Fig 1.** Landmarks used for the measurement of viscoelasticity in the present study (A), measurement of viscoelasticity in the facial soft tissue with a viscoelasticity measuring instrument (B), and the indenter of the viscoelasticity measuring instrument (C). Chk, cheek; Cphs, crista philtri superior, Cphs', the point on the philtral crest approximately 12 mm below the Cphs (newly defined); Cphi, crista philtri inferior; Ch, cheilion; L, cleft side; R, right side. Black arrow in (B): probe of the viscoelasticity measuring instrument (Vesmeter-E100Hs; WaveCyber Corp., Saitama, Japan). Black arrow in (C): indenter of the viscoelasticity measuring instrument.

standardized based on our previous study [19]. In brief, we used the common coordinate system based on 19 square regions (size, 4.0 × 4.0 mm; each square included 81 points) located on the forehead and the nasal bridge [19]. These areas were assumed to be the regions that remained immobile while smiling. For the standardization process, we employed the Iterative Closest Point algorithm. The mean anteroposterior distance between 2 facial positions for the 19 square regions was 0.26 mm, which was considered sufficiently small for subsequent calculation.

To standardize the differences in facial size, the images were normalized by the distance between the right and left exocanthions. Mirror images of the patients with right cleft lips were mathematically created to produce images for patients with left cleft lips.

## Wire mesh fitting

For each participant, 3D faces were fitted with mathematical wire meshes based on the landmarks shown in Fig 2 using the Face-Rugle software program (Ver. 1.01; Medic Engineering

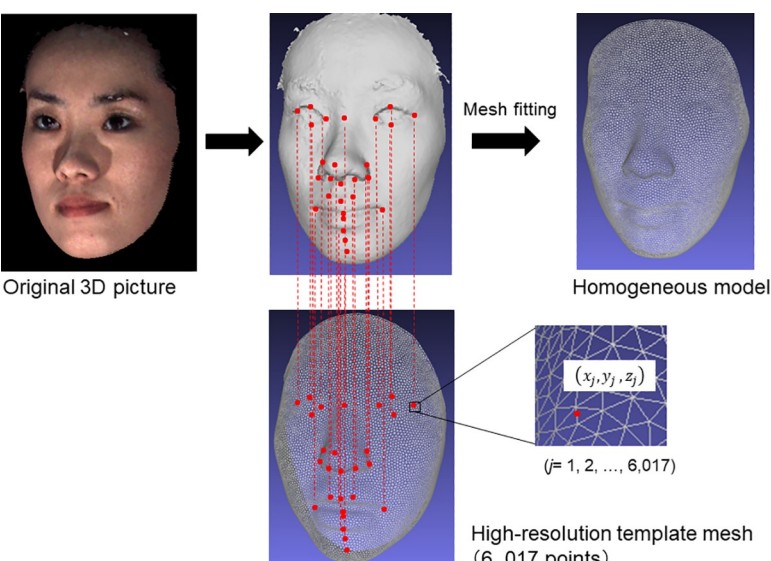

**Fig 2. Three-dimensional image of the mesh fitting method.** For each facial model, template meshes were fit using a software program, based on landmarks. This method automatically generated a point cloud, creating a set of 6,194 data points in a 3D coordinate system for each facial expression.

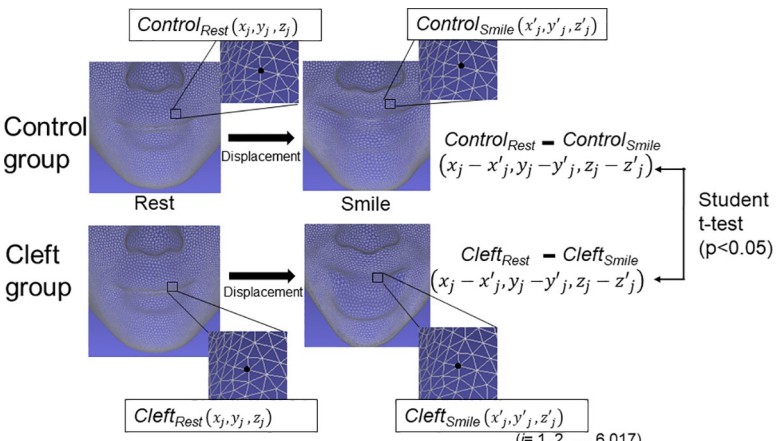

**Fig 3. Schematic illustration of the statistical comparison of the displacements from a rest position to the peak maximum smile position between the Control group and Cleft group.** $Control_{rest}$, faces at rest in the Control group; $Control_{smile}$, faces at the peak of smiling in the Control group; $Cleft_{rest}$, faces at rest in the Cleft group; $Cleft_{smile}$, faces at the peak of smiling in the Cleft group.

Co., Kyoto, Japan), and the nodes of the fitted meshes (semi-landmarks) were used in the statistical analyses below. This method generated a point cloud, which was a set of 6,194 semi-landmark data points in a 3D coordinate system for each facial expression (Fig 2) [19, 22].

## Examination of the displacement from the rest to peak maximum smile

To determine displacement in the soft tissue surface morphology from rest to the maximum smile, the displacement of each point on the facial surface in the X, Y, and Z axes between the two facial positions was calculated. A two-sample *t*-test was performed to compare the displacement between two subject groups in each axis with adjustments for multiple comparisons by the Benjamini-Hochberg method (Fig 3). To visualize differences in displacement between the two subject groups, the results were represented as a color map showing the p-values for the aforementioned comparison between the two subject groups (significance probability map) and a color map showing the differences in the displacements while smiling between the two subject groups (distance map) [19, 22–24]. The statistically significant level was set at P < 0.05.

Furthermore, in order to examine the main effects in displacements in the X-, Y-, and Z-axes, we performed linear regression analyses where X-, Y-, Z displacement values were set as the dependent variables and the Cleft group was assigned a value of 1 and the Control group was assigned a value of -1 as a response variable (S2 Fig). The results were also represented as three types of color maps showing 1) the p-values of the linear regression model, 2) the estimated coefficients for the X-, Y-, Z displacements of the model (coefficient map), and 3) the p-value for the coefficient value. Adjustments for multiple comparisons were conducted by the Benjamini-Hochberg method.

## Variance of each group, each sex, and their interactions

To examine the variance in each group, each sex, and their interactions, two analyses were conducted using a method reported previously [25, 26]. We first performed a principal component analysis (PCA) for the 6,194 coordinates of the displacement of each point on the facial surface in the X, Y, and Z axes between the two facial positions to reduce dimensionality. Significant principal components (PCs) were determined by a scree plot analysis and entered into a MANOVA to test for significance of the factors, i.e. cleft/non-cleft and sex. If there was a

statistical association between a dependent vector (i.e. PCs to represent facial displacement while smiling) and independent grouping variable of cleft/non-cleft, PC regression was conducted to examine the degree of contribution of each PC, where the Cleft group was assigned a value of 1 and the Control group was -1 as a response variable.

Furthermore, the determined PCs were used to detect relationships between facial displacement while smiling and physical properties of the skin, as described in detail later in this manuscript.

## Physical properties

A two-sample *t*-test was used to examine any significant differences in elastic modulus and the viscosity coefficient between the Cleft group and Control group. A normal data distribution was statistically confirmed using the Kolmogorov-Smirnov test. When the mean differences between groups showed values exceeding the 95% confidence level ($MDC_{95}$) defined in S1 Text and S1 Table [27] and P-values less than 0.05 were observed, we confirmed a statistically significant difference. Adjustment for multiple comparisons was conducted using the Benjamini-Hochberg method. To include any associations between landmarks and viscosity coefficient and elastic modulus, we performed linear regression analyses where viscosity coefficient and elastic modulus of the four paired landmarks were set as the dependent variables and the Cleft group was assigned a value of 1 and the Control group was assigned a value of -1 as a response variable.

## Relationship between facial displacement while smiling and physical properties of the skin

To examine the direct relationship between facial displacement and the physical properties of the skin, a canonical correlation analysis (CCA) was conducted. A CCA is a multivariate exploratory approach that highlights correlations between two sets of variables. For the physical properties, 16 parameters of 2 physical properties (elastic modulus and the viscosity coefficient) at 4 landmarks (*Chk*, *Cphs'*, *Cphi*, and *Ch*) on both sides were employed. For the facial displacement while smiling, the PCs derived from facial displacement (described in the section above) were employed.

## Craniofacial characteristics of the samples

To obtain baseline information on the craniofacial characteristics of the patients in the Cleft group, cephalometric radiographs of the patients taken within six months of the time point when 3D facial images had been taken were obtained, digitized, traced, and analyzed by one experimenter (D.L.) using a commercial software program (Dolphin Imaging 11.0; Dolphin Imaging & Management Solutions, Chatsworth, CA, USA) for the 13 cephalometric parameters (for parameter definitions, please see S2 Text and S2 Table [28]).

## Results

The cephalometric analysis revealed that our samples were considered to have normal (Class I) skeletal relationships (S1 Table).

## Displacement differences while smiling between the Cleft group and Control group

Figs 4 and 5 show the results of the comparisons between the Cleft group and the Control group for the displacement from rest to peak smile. There were significant differences between

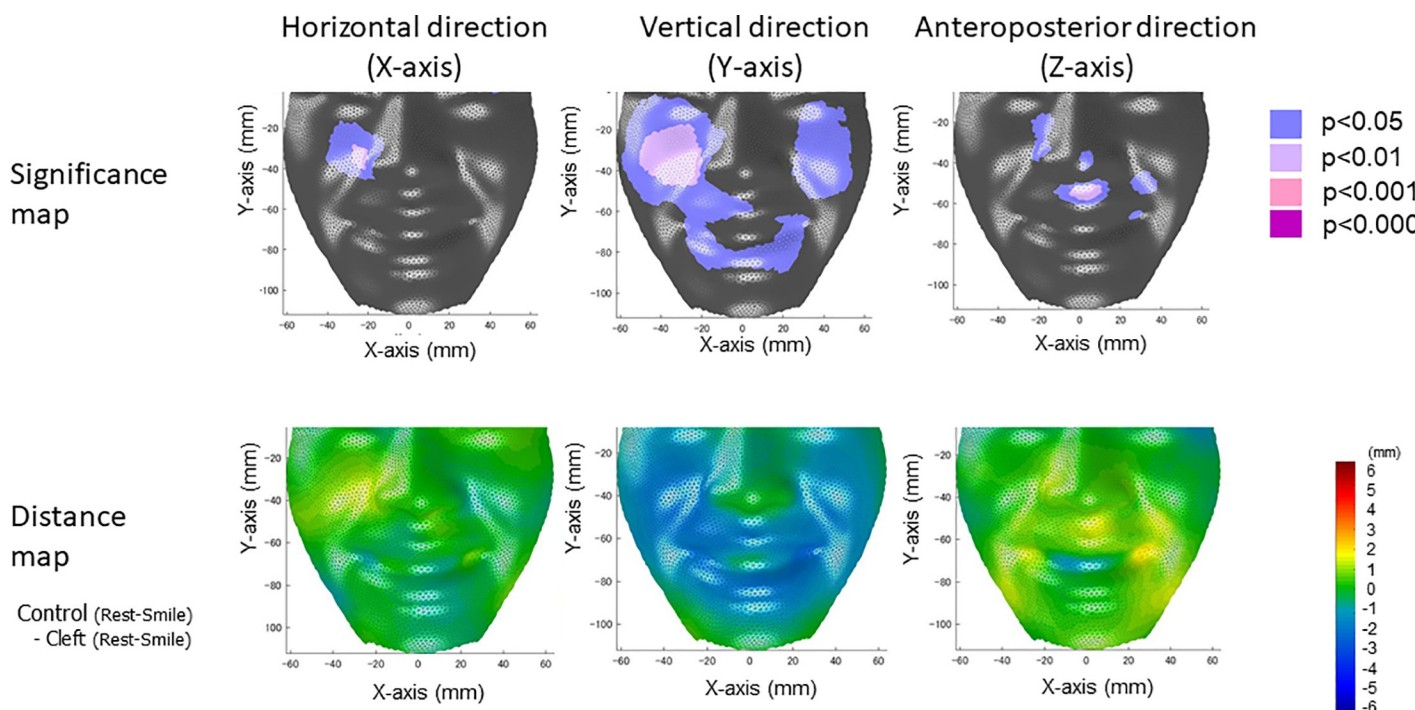

**Fig 4. Significance probability maps (top) and distance maps (bottom) of the difference in displacement while smiling between the Control Group and the Cleft group in three directions (horizontal [left], vertical [middle], and anteroposterior directions [right]).** For the significance probability maps, blue indicates P < 0.05; pale pink, P < 0.01; dark pink, P < 0.001; purple, P < 0.0001. Adjustment for multiple comparisons was conducted using the Benjamini-Hochberg method. For the distance maps, yellow indicates that the difference in the displacement between the two subject groups (Control group–Cleft group) is a positive value, whereas blue indicates that the difference is a negative value. Displacement was defined as differences in the coordinate values of the face at rest minus those at the peak of the smiling. Thus, in the horizontal direction, on the non-cleft side (right face), yellow indicates a smaller lateral displacement or greater medial movement while smiling in the Cleft group than in the Control group, whereas blue indicates a greater lateral displacement or a smaller medial movement while smiling than in the Control group. On the cleft side (left face), yellow indicates a greater lateral displacement or smaller medial movement while smiling in the Cleft group than in the Control group, whereas blue in the face indicates a smaller lateral displacement or a greater medial movement while smiling than in the Control group. In the vertical direction, blue indicates a greater downward displacement or a smaller upward displacement while smiling in the Cleft group than in the Control group. In the anteroposterior direction in the Cleft group, yellow indicates a smaller retrusive movement or a greater protrusive movement, whereas blue indicates a greater retrusive movement or a smaller protrusive movement than in the Control group.

the two sample groups in the cheek, nasal dorsum, alar, subalar, upper and lower lips, labial commissure, and chin. The results for each facial region are summarized in Tables 2 and 3. In brief, on both sides, the upward and backward displacement of the upper lip and the labial commissure while smiling was smaller in the Cleft group than in the Control group. In contrast, the downward displacement of the lower lip was greater in the Cleft group than in the Control group. Lateral subalar displacement on the non-cleft side while smiling was smaller in the Cleft group than in the Control group. When considering interactions among three axes using a regression model (S3 Fig), the overall results coincided with the results of the statistical comparison in each axis using the *t*-test (Fig 4). However, the aforementioned smaller lateral displacement of the subalar area on the non-cleft side was found to be due to a greater downward movement of the cheek on the non-cleft side in the Cleft group than in the Control group. Further, increased horizontal movements of the corner of the mouth on both sides and the subalar area on the cleft side was also found to be related to discrimination of the subject groups. These results suggest that the lateral movement of the nasal ala was restricted on the non-cleft side while flexibility of the nasal ala on the cleft side was increased in the Cleft group.

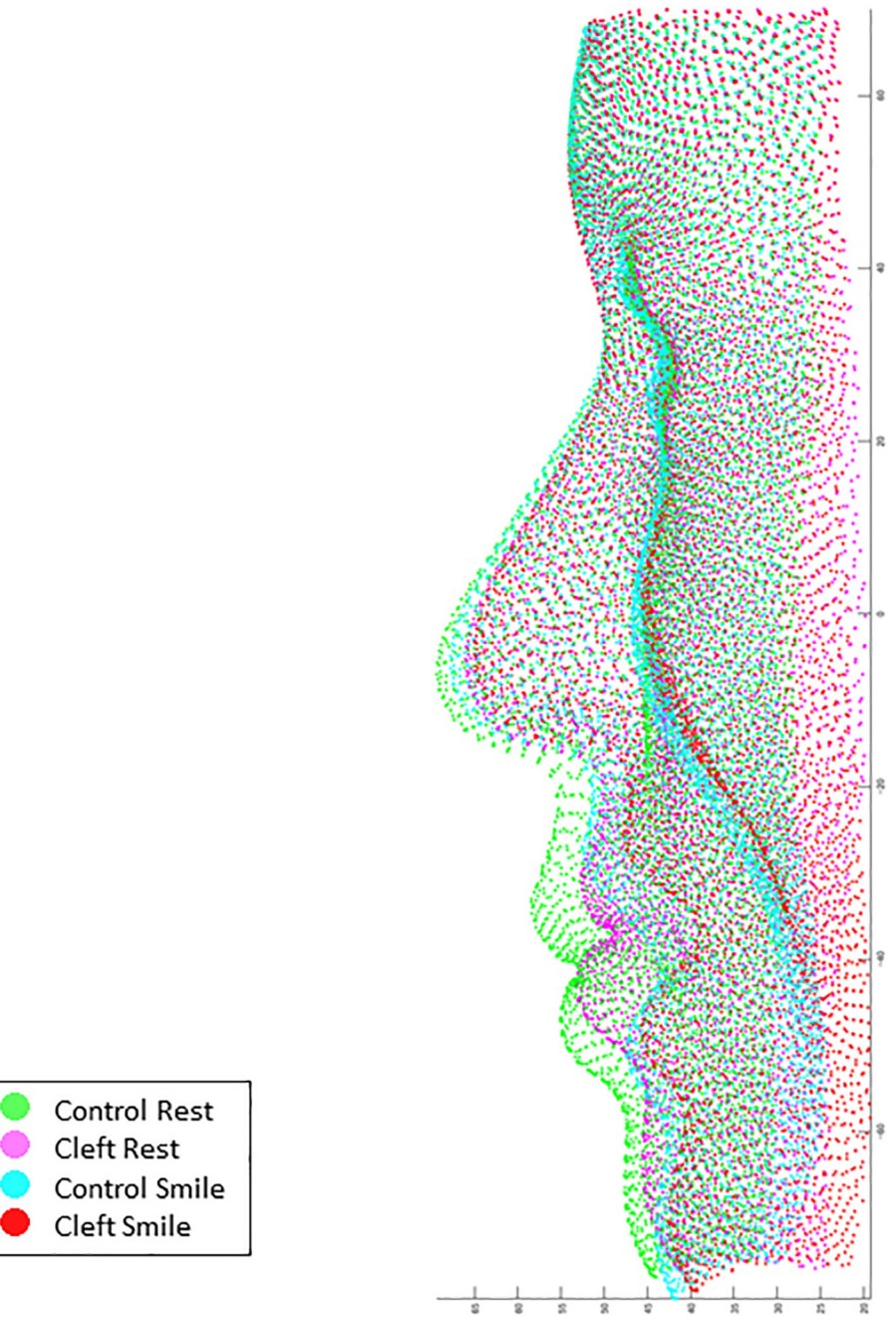

**Fig 5.** Superimposition of the average coordinate values of the Control group at rest posture (green), the Cleft group at rest posture (pink), the Control group at the peak of smiling (cyan), and the Cleft group at the peak of smiling (red).

## Variance of each cleft/non-cleft, each sex, and their interactions

Similar findings to the above results were holistically found in the PC regression analysis (Fig 6). The first 10 significant PCs, which explained 78.3% of the sample's variance, were determined to be significant by a scree plot analysis to express the facial displacement while smiling. Principal component regression showed that PC 1 (weight = 0.30), PC 2 (0.18), PC 6 (-0.23), PC 8 (0.19) and PC 9 (0.18) had a significant effect on the Cleft group (p = 0.018, $R^2$ = 0.25).

**Table 2. Summary of the displacement while smiling in the two groups on the non-cleft side.**

| Area | Lateral–displacement (X-value) | Upward—displacement (Y-value) | Downward—displacement (Y-value) | Backward–displacement (Z-value) |
|---|---|---|---|---|
| **Cheek** | **Cleft < Control (sub-effect to †^) | *Cleft < Control (†) | | ** Cleft < Control |
| **Nasal—dorsum** | NS | NS | NS | NS |
| **Alar** | **Cleft < Control (sub-effect to ¶^) | **Cleft < Control (¶) | | *Cleft < Control |
| **Subalar** | NS | NS | NS | NS |
| **Upper lip** | NS | *Cleft < Control | NS | **Cleft < Control |
| **The labial commissure** | Cleft > Control^ | *Cleft < Control | | *Cleft < Control |
| **Lower lip** | NS | | *Cleft>Control | NS |

** P < 0.01

* P < 0.05; NS, not significant

^, results of regression analysis.

Cleft > Control, the Cleft group showed a significantly greater average displacement than the Control group; Cleft < Control, the Control group showed a significantly greater average displacement than the Cleft group.

PC 1 was characterized by the amount of displacement of the facial surface while smiling, and the Cleft group showed smaller displacement than the Control group (p < 0.01). PC 2 was characterized by vertical movement of the lower lip, and the Cleft group showed a tendency toward smaller values of PC 2 than the Control group, indicating a tendency towards downward movement of the lower lip. PC 6 was characterized by less cheek upward movement while smiling, and the Cleft group showed a tendency toward chin protrusion while smiling and less cleft-side cheek upward movement than the Control group. PC 8 was characterized by asymmetric movement, including greater lateral displacement of the subalar on the cleft side while smiling and greater upward movement of the corner of the mouth on the cleft side than on the non-cleft side. These results appear to be consistent with those from the significance probability map, which indicated that lateral movement of the nasal alar was restricted on the non-cleft side while flexibility of the nasal alar on the cleft side was increased in the Cleft

**Table 3. Summary of the displacement while smiling on the cleft side in the two groups.**

| Area | Lateral–displacement (X-value) | Upward–displacement (Y-value) | Downward–displacement (Y-value) | Backward–displacement (Z-value) |
|---|---|---|---|---|
| **Cheek** | NS | * Cleft < Control | | NS |
| **Nasal—dorsum** | NS | NS | NS | NS |
| **Alar** | NS | NS | NS | *Cleft < Control |
| **Subalar** | Cleft > Control^ | NS | NS | *Cleft < Control |
| **Upper lip** | NS | *Cleft < Control | | **Cleft < Control |
| **The labial commissure** | Cleft > Control^ | *Cleft < Control | | **Cleft < Control |
| **Lower lip** | NS | | *Cleft > Control | NS |

** P < 0.01

* P < 0.05; NS, not significant

^, results of regression analysis.

Cleft > Control, the Cleft group showed a significantly greater average displacement than the Control group; Cleft < Control, the Control group showed a significantly greater average displacement than the Cleft group.

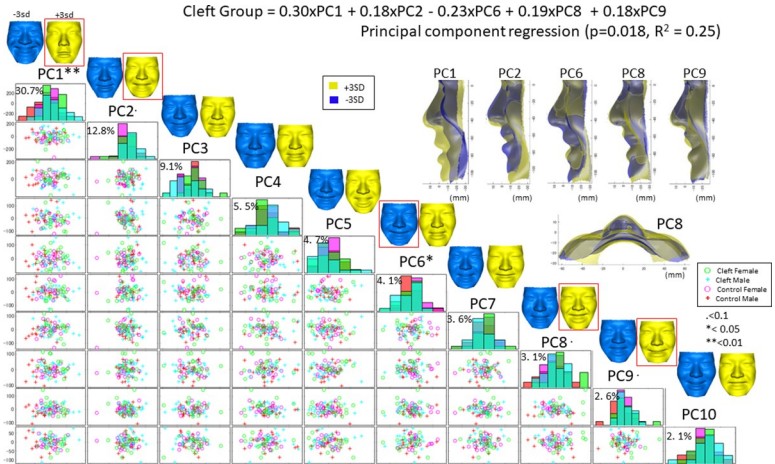

**Fig 6. A scatter plot matrix of the principal component (PC) scores in the Cleft and Control groups with a histogram in the diagonal cells.** PCs 1–10 explain 78.3% of the shape variation across samples. Green denotes facial configurations in the female Cleft group, cyan denotes those in the male Cleft group, pink denotes those in the female Control group, and red denotes those in the male Control group. The PCs are defined as +1 SD (yellow) and −1 SD (blue) in the top column. For example, the top column indicates that a +3 SD value of PC1 represents a smaller facial displacement while smiling, shown in yellow (+3 SD). The histogram shows that the bimodal distribution and patients in the Cleft group (cyan) showed a greater PC1 value than the Control group, suggesting that the Cleft group had a smaller facial displacement while smiling. The red squares show PCs that were representative of the Cleft group based on the PC regression analysis (P<0.1).

group. PC 9 was characterized by less upward and backward movement of the upper lip in the Cleft group than in the Control group.

MANOVA showed that sex and the cleft are the significant factors that affect the facial displacement while smiling (P<0.01, Table 4), but no significant interaction was found for these two factors (p = 0.80). This result indicates that the displacement of the facial surfaces from rest to peak smiling was significantly different between the Cleft and Control groups and sex subgroups; however, the effects of these two factors (sex and cleft) on displacement while smiling were independent.

## Physical properties of the skin

The normal distribution of the physical properties in each group was clarified. The Cleft group showed a significantly greater elastic modulus than the Control group at *Cphs'* on the cleft

**Table 4. Results of a multifactor analysis of variance.**

| | Df | Pillai | Approx F | Num Df | Den Df | Pr(>F) |
|---|---|---|---|---|---|---|
| **Cleft** | 1 | 0.2522 | 2.327 | 10 | 69 | 0.020* |
| **Sex** | 1 | 0.2977 | 2.926 | 10 | 69 | 0.004* |
| **Cleft:Sex** | 1 | 0.0945 | 0.720 | 10 | 69 | 0.702 |
| **Residuals** | 78 | | | | | |

* P<0.01.

Df, degrees of freedom; Pillai, Pillai's trace (a test statistic in the multifactor analysis of variance; this is a positive-value statistic ranging from 0 to 1, and increasing values mean that the effects are contributing more to the model); Approx F, the F statistic for the given predictor and test statistic; Num DF, number of degrees of freedom in the model; Den DF, number of degrees of freedom associated with the model errors; Pr(>F), p-value associated with the F statistic of a given effect and test statistic (the null hypothesis that a given predictor has no effect on either of the outcomes is evaluated with regard to this p-value).

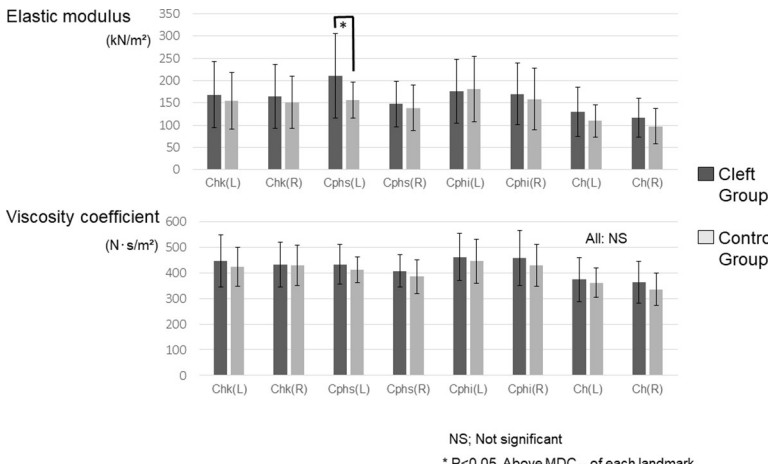

**Fig 7. A statistical comparison of the elastic modulus (top) viscosity coefficient (bottom) between the Cleft group and the Control group.** Chk, cheek; Cphs', crista philtri superior'; Cphi, crista philtri inferior; Ch, cheilion; NS, not significant. MDC95: minimum detectable change (calculated in S1 Text). Adjustment for multiple comparisons was conducted using the Benjamini-Hochberg method.

side, indicating stiffer characteristics of the scar than normal skin (P < 0.05) (Fig 7). The standard deviation of the elastic modulus was greater in the Cleft group than in the Control group at all landmarks except for *Cphi* on both sides. No significant difference in the viscosity coefficient was observed between the groups (Fig 7.). Similarly, only the greater elastic modulus at *Cphs'* on the cleft side was found to be the main effect in the Cleft group (S3 Table).

## Relationship between facial displacement while smiling and physical properties of the skin

The CCA analysis revealed a significant association between facial displacement while smiling and physical properties of the skin (correlation coefficient = 0.82, p = 0.04, Wilks' Lambda = 0.03, F = 1.38, df1 = 80, df 2 = 154.4). The heatmap of the correlation matrix is shown in Fig 8. The elasticity of the Cphs (cleft side), the viscosity and elasticity of the Ch (cleft side), and the elasticity of the Cphi (non-cleft) were found to be positively related to PCs 2 and 9 of facial displacement while smiling. This indicates that hard skin at the scar of the repaired cleft and labial commissure on the cleft side was associated with greater lower lip downward movement and less upward upper lip movement while smiling. In contrast, greater asymmetric displacement (represented by PC8) and less cheek upward movement (represented by PC 6) showed relatively little association with the physical characteristics of the skin.

## Discussion

To our knowledge, this is the first study to evaluate the relationship between the physical property of the scar with surrounding soft tissue and facial displacement in patients with UCLP. Our results regarding the displacement of facial soft tissue while smiling revealed that patients with UCLP showed interrupted displacement of facial soft tissues while smiling. We also found greater viscoelasticity of the scar and surrounding facial soft tissue in patients with UCLP than in healthy adults. Finally, we clarified the significant relationship between viscoelasticity of the scar and surrounding facial soft tissue and displacement of facial soft tissue while smiling in patients with UCLP.

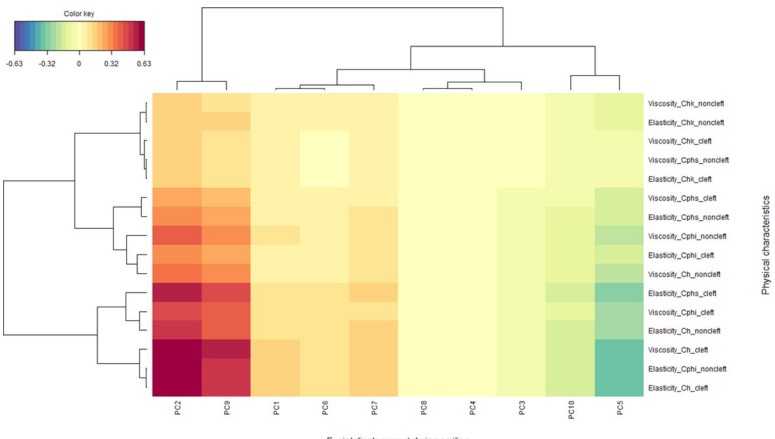

**Fig 8. Heatmap showing the correlation between the facial displacement while smiling, represented as the principal component (PC), and the physical characteristics of the skin according to a canonical correlation analysis.** The X-axis depicts the 3D facial displacement while smiling represented as PCs 1–10 and the Y-axis the viscoelasticity of the skins tested. The red and blue colors indicate strong positive and negative correlations, respectively, whereas yellow indicates weaker correlation values (see color key). The "mixOmics" tool as part of the R software package was used to draw this figure.

## Displacement of facial soft tissue while smiling

As typical displacements of the facial soft tissue of the Cleft group, there are three noted movement patterns: reduced upward and backward displacement in the scar area, downward movement of the lower lip, and increased asymmetric lateral displacement. Each facial movement while smiling specific to the Cleft group is discussed below.

## Reduced upward and backward displacements on the scar area

The upward and backward displacement of the upper lip and the labial commissure while smiling was found to be smaller in the Cleft group than in the Control group on both sides. Furthermore, restricted anteroposterior and vertical motions were clearly observed at the scar using significance probability maps. A PCA showed that this restricted displacement of the upper lip was also observed as a reduced dimension of PC 9. An examination of the relationship between the facial displacement and the physical property using a CCA showed that PC 9 was significantly associated with increased viscoelasticity of the scar and surrounding facial soft tissue. Furthermore, we found a greater elastic modulus (hardened soft tissue) in the Cleft group than in the Control group at the crista philtri superior on the cleft side, corresponding to the scar of the repaired cleft in the Cleft group. Based on these results, the restricted movement of the upper lip was attributed to the hardened soft tissue corresponding to the scar tissue. This is consistent with the fact that scar formation is the product of collagen fiber synthesis and alignment in the presence of a tensile stress field generated by the wound contraction process for a prior cleft [8].

In addition, because we evaluated patients with normal occlusion and normal cranio-facial characteristics, the effect of skeletal 3 malocclusion on facial displacement while smiling was excluded from the causative factors of abnormal facial displacement in the present study. We can therefore say that orthodontics alone cannot improve this restricted movement of the upper lip. Thus, while assumptions have been proposed in several studies of the restricted movement of the upper lip, this is the first report to prove the relationship between hardened scar tissue and reduced upper lip movements while smiling.

## Downward movement of the lower lip

We also found a significant downward displacement in the lower lip and chin areas (PC 2) in relation with the increased viscoelasticity of the scar and surrounding facial soft tissue. A previous study indicated that the downward displacement of the lower lip while smiling was related to the displacement of the lower jaw [29]. However, in the present study, participants were instructed to smile with their teeth in light contact in the habitual maximum intercuspation position. Therefore, we speculated that the movement of the lower jaw did not affect the displacement of the lower lip area. The downward displacement of the lower lip while smiling in the Cleft group was thus concluded to be a compensatory movement for reduced movement of the upper lip while smiling.

## Increased asymmetric displacement while smiling

In the present study, the Cleft group showed two main features of asymmetric movement: greater lateral displacement of the subalar on the cleft side while smiling and greater alar backward displacement on the non-cleft side (i.e. flattening of the nasal alar at the non-cleft side). This was expressed mainly as PC 8, and the results showed that the nasal alar was restricted on the non-cleft side, while flexibility of the nasal alar on the cleft side was increased in patients. Interestingly, this result on the cleft side was the opposite of the displacement that had been initially expected, where the tightened cleft scarring on the upper lip seemed to pull the nose and surrounding upper lip area toward the scar during smiling on both sides. Furthermore, the CCA revealed that these asymmetric characteristics were not closely associated with the physical characteristics of the skin. These results suggest that the tightened cleft scarring on the upper lip was unlikely to be related to the asymmetric displacement of the face while smiling, hinting at other causative factors for this asymmetric displacement of the nose and upper lip while smiling.

There are two possible causative factors for the asymmetric movement of the nose and upper lip: First, the two functional components of the orbicularis oris muscle (i.e. extrinsic and intrinsic bundles), which function as the retractor and constrictor of the mouth, respectively, were considered to be impaired even after the first repair surgery. A previous report [30] showed that the extrinsic bundles of the orbicularis oris muscle were displaced and the muscle's direction changed by the cleft in patients with unilateral cleft lip. The extrinsic bundles are the retractors of the orbicularis oris muscle, and disfigurement of the extrinsic bundles may result in the asymmetric backward displacement of the scar area while smiling. In addition, an impaired muscle function can interrupt normal displacement of facial soft tissues. Second, an impaired perioral sensorimotor system in patients with UCLP can affect the displacement of facial soft tissues. A normal sensory function is necessary for the normal motor function of the perioral area [31, 32]. A previous study assessed altered sensation areas by an interview and showed the existence of an abnormal neurosensory function around the perioral area in patients with CLP [7]. An evaluation of the upper lip area by electromyography also revealed the asymmetric distribution of the muscle activity around the cleft area, indicating an impaired motor unit around the cleft [33]. Impaired neurosensory and motor functions in patients with UCLP may disturb the displacement of the perioral area while smiling.

## Clinical applications

Based on these results, the tightened scarring is considered to be related to abnormal facial displacement while smiling, i.e. less upward and backward displacements on the scar area and downward movement of the lower lip. However, the asymmetric displacement (i.e. restricted nasal movement on the non-cleft side) and increased flexibility of the subalar on the cleft side

are expected to have causative factors other than scarring. Therefore, to improve this abnormal movement while smiling, these causative factors should be separately considered and addressed.

For example, in clinics, if hardened scar tissue restricts the upper lip movement and reduces cheek and nasal alar movement on the non-cleft side in a patient, various treatment options to soften the scar can be suggested, including Er:YAG laser [34], silicone gel sheet [35], and massage therapy [36]. However, if hardened scar tissue does not restrict movement, cheiloplasty with z-plasty can be suggested. If downward movement of the lower lip while smiling is observed in a patient, smiling training after improvement of the upper lip movement may be suggested. If increased movement of the nasal alar on the cleft side is observed, reconstruction of the orbicularis oris muscle may be suggested. The best treatment option for improving the facial motion function should be determined by combining examinations of the 3D facial morphology and physical properties of facial soft tissues in clinical settings.

### Limitations

Several limitations associated with the present study warrant mention. First, the sample size for the analysis of the effect on facial displacement based on skeletal parameters was relatively small; therefore, our results might have differed had we had access to a larger sample. Second, our sample population had little variation in age (15–37 years) and ethnicity (Japanese) and was from one center (Osaka University Dental Hospital). Therefore, readers should consider these sample variations when applying our findings to a different population and different center. Third, the present study was only limited to assessing range of motion limitation during the raising of the lips. Apparently, further studies are needed to cover multiple facial expressions (e.g., lip protrusive movement [as an anterior motion limitation]) to improve patients' social function of non-verbal communications.

### Conclusion

There are three typical abnormal features of smiling in patients with cleft lip and palate: (1) less upward and backward displacement on the scar area, (2) downward movement of the lower lip, and (3) greater asymmetric lateral displacement, including greater lateral displacement of the subalar on the cleft side while smiling and greater alar backward displacement on the non-cleft side. Physical properties of the scar and surrounding facial soft tissue were found to affect features (1) and (2) of the facial soft tissue while smiling in patients with repaired UCLP; however, asymmetric feature (3) seems to be caused by factors other than hardened soft tissue at the scar. Thus, in the clinical setting, it is important to examine the 3D facial morphology and physical properties of facial soft tissue in order to determine the optimum treatment options to correct functional impairment in facial expression generation in patients with repaired unilateral cleft lip.

### Supporting information

**S1 Fig. Three-dimensional coordinate system.** The nasion (N) was defined as the origin (O). The sagittal plane was defined as the plane passing through the origin and perpendicular to the line through the midpoint of the right exocanthion (Ex) and endocanthion (En) and the midpoint of the left Ex and En. The axial plane was defined as the plane passing through the origin and parallel to the line connecting the porion and geometric center (g) of the porion (Po), subnasale (Sn), and Ex on the image projected onto the sagittal reference plane. The coronal plane was defined as a plane passing through the origin and perpendicular to both the axial and sagittal planes. + indicates the positive direction in each axis (Cited from Tanikawa et al. [22] with

permission).
(TIF)

**S2 Fig. Schematic illustration of the statistical analyses to examine main effect in displacement from a rest position to the peak maximum smile position in X-, Y-, Z-axes.** For each semi-landmark of the facial surface, linear regression was applied, where the X-, Y-, Z-displacement values were set as the dependent variables and the Cleft group was assigned a value of 1 and the Control group was assigned a value of -1 as a response variable. The *cx*, *cy*, and *cz* were coefficients for the regression model.
(TIF)

**S3 Fig. Significance probability maps (top) and coefficient maps (bottom) of the linear regression, where the X-, Y-, Z-displacement values were set as the dependent variables and the Cleft group was assigned a value of 1 and the Control group was assigned a value of -1 as a response variable.** (Model significance [left], horizontal [second left], vertical [second right], and anteroposterior directions [right]). For the significance probability maps, blue indicates $P < 0.05$; pale pink, $P < 0.01$; dark pink, $P < 0.001$; purple, $P < 0.0001$. Adjustment for multiple comparisons was conducted using the Benjamini-Hochberg method. For the coefficient maps, red and yellow indicate that the coefficient is a positive value, whereas blue indicates that coefficient is a negative value. Displacement was defined as differences in the coordinate values of the face at rest minus those at the peak of smiling. Thus, in the horizontal direction, yellow indicates that the difference in the displacement between the two subject groups (Control group–Cleft group) is a positive value, whereas blue indicates that the difference is a negative value. Displacement was defined as the difference in the coordinate values of the face at rest minus those at the peak of smiling. Thus, in the horizontal direction, on the non-cleft side (right face), yellow indicates a smaller lateral displacement or greater medial movement while smiling in the Cleft group in comparison to the Control group, whereas blue indicates a greater lateral displacement or a smaller medial movement while smiling in comparison to the Control group. On the cleft side (left face), yellow indicates a greater lateral displacement or smaller medial movement while smiling in the Cleft group in comparison to the Control group, whereas blue in the face indicates a smaller lateral displacement or a greater medial movement while smiling in comparison to the Control group. In the vertical direction, blue indicates that a greater downward displacement or a smaller upward displacement while smiling is related to the Cleft group. In the anteroposterior direction in the Cleft group, red and yellow indicates that a smaller retrusive movement or a greater protrusive movement is related to the Cleft group, whereas blue indicates that a greater retrusive movement or a smaller protrusive movement is related to the Cleft group.
(TIF)

**S1 Table. Mean and standard deviation (SD) of the minimal detectable change for each landmark at the 95% confidence level (MDC$_{95}$) between Session 1 and Session 2.**
(DOCX)

**S2 Table. Dental and skeletal parameters of the Cleft group.**
(DOCX)

**S3 Table. Dental and skeletal parameters of the Cleft group.**
(DOCX)

**S1 Text. Intermediate precision of measuring devices for physical properties of facial landmarks.**
(DOCX)

**S2 Text. Craniofacial characteristics of the patients in the Cleft group.**
(DOCX)

## Author Contributions

**Conceptualization:** Chihiro Tanikawa.

**Data curation:** Donghoon Lee, Chihiro Tanikawa.

**Formal analysis:** Donghoon Lee, Chihiro Tanikawa.

**Funding acquisition:** Chihiro Tanikawa.

**Investigation:** Donghoon Lee, Chihiro Tanikawa.

**Methodology:** Chihiro Tanikawa.

**Project administration:** Chihiro Tanikawa.

**Resources:** Chihiro Tanikawa.

**Software:** Chihiro Tanikawa.

**Supervision:** Chihiro Tanikawa.

**Validation:** Chihiro Tanikawa.

**Visualization:** Chihiro Tanikawa.

**Writing – original draft:** Donghoon Lee, Chihiro Tanikawa.

**Writing – review & editing:** Chihiro Tanikawa, Takashi Yamashiro.

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
