## [Decision Letter · Decision Letter 0]

3 Feb 2021

PONE-D-20-33973

Functional impairment in facial expression generation in patients with repaired unilateral cleft lip: Effects of the physical properties of facial soft tissues

PLOS ONE

Dear Dr. Tanikawa,

Thank you for submitting your manuscript to PLOS ONE. After careful consideration, we feel that it has merit but does not fully meet PLOS ONE’s publication criteria as it currently stands. Therefore, we invite you to submit a revised version of the manuscript that addresses the points raised during the review process.

Although the reviewers’ comments are short, they are critically important. The reviewers and I are not 100% convinced about the correctness of the statistical analysis. Please, address the reviewers’ comments on statistical analyses and, if necessary, expand the explanations to support their use. I would like to give you the opportunity to check that issue and explain yourself in the rebuttal letter; however, note that if the doubts about statistics persist after review, I may need consult with additional reviewers or with the PLOS ONE statistics support to make sure the analysis is sound.

Reviewers also disagree on data availability, please check if you comply with PLOS ONE policy on data sharing.

Finally, take into consideration the suggestions of Reviewer 2 on extending basic explanations in the main text that are now too succinct. Think that PLOS ONE is a generalist outlet with a broad audience.

We look forward to receiving your revised manuscript.

Kind regards,

Borja Esteve-Altava, Ph.D.

Academic Editor

PLOS ONE

Journal Requirements:

2. Figures Include Patient Photos – Patient Consent for Publication

We note that Figures S1, S2 and S3 and Figures 1 & 2 include images of participants in the study.

As per the PLOS ONE policy (http://journals.plos.org/plosone/s/submission-guidelines#loc-human-subjects-

research) on papers that include identifying, or potentially identifying, information, the individual(s) or

parent(s)/guardian(s) must be informed of the terms of the PLOS open-access (CC-BY) license and provide

specific permission for publication of these details under the terms of this license.

Please download the Consent Form for Publication in a PLOS Journal (http://journals.plos.org/plosone/s/file?id=8ce6/plos-consent-form-english.pdf). The signed consent form should not be submitted with the manuscript, but should be securely filed in the individual's case notes.

Please amend the methods section and ethics statement of the manuscript to explicitly state that the patient/participant has provided consent for publication: “The individual in this manuscript has given written informed consent (as outlined in PLOS consent form) to publish these case details”.

If you are unable to obtain consent from the subject of the photographs, you will need to remove the figures

and any other textual identifying information or case descriptions for these individuals.

3. You indicated that the inclusion criteria for participants within your study is 15–37 years old.

In your Methods section, please ensure you have also stated whether you obtained consent from parents or guardians of the minors included in the study or whether the research ethics committee or IRB specifically waived the need for their consent.

Reviewers' comments:

Reviewer's Responses to Questions

**Comments to the Author**

1. Is the manuscript technically sound, and do the data support the conclusions?

Reviewer #1: Yes

Reviewer #2: Yes

2. Has the statistical analysis been performed appropriately and rigorously? 

Reviewer #1: I Don't Know

Reviewer #2: No

3. Have the authors made all data underlying the findings in their manuscript fully available?

Reviewer #1: Yes

Reviewer #2: No

4. Is the manuscript presented in an intelligible fashion and written in standard English?

Reviewer #1: Yes

Reviewer #2: No

5. Review Comments to the Author

Reviewer #1: This paper were analyzed facial morphology while smiling in patient with unilateral cleft lip and palate. Correlation between facial displacement and physical property is interesting and should be beneficial information for readers of the journal. However, there is a room for improvement as follows;

Title, Authors started the title with Functional impairment. However, the movement mentioned in this study was just for smiling and at rest position. The movement written in this study technically was not the function. Please reconsider about the title.

P9L2; Authors recorded each expression once. Reproducibility of each position was not mentioned even if participants were trained several times. Reproducibility should be added.

P14L1-2; Expression should be revise.

P40L6; Facial expression evaluated in this study was only for the maximum smiling. However, there should be more expressions other than maximum smiling. It would be too hasty to reach the clinical options including surgery from these results. Please reconsider or add the discussion about this point.

Reviewer #2: This is a clinical experiment where they test that physical properties of the scar and surrounding facial soft tissue might affect facial displacement while smiling in patients with UCLP. I find the work interesting and the research well performed but I think that needs a better writing. I am familiar with clinical journals and I know that most of the works are written systematically, but PLoS ONe is not a clinical journal and, therefore, some of the text needs a better explanation.

For example, you cannot write "For details, please see our previous study". Instead, you need to explain a summary of the method and cite the study. The manuscript itself should explain all the details of the research to make it repeatable for others. The methods part needs to be more explained. I discovered that to obtain the Elastic and Viscosity parameters you were using a device because it is explained in the legend of the figure. This should be explained in the core text. Moreover, I went to other publication to guess that elastic and viscoelastic parameters were based in the Voigt equation. This should be explained briefly in the text, too. And cite the reference as well. Moreover, the method used to compute the Elastic and Viscosity parameters using the Vesmeter device is defined in one reference, but you should -at least- explain the basics in these documents. Especially because the information you can find in Internet provided by the manufacturer of this device is only in Japanese.

I have a second concern about the results and their statistical treatment. For example, I am not sure that a two-sample t-test can be performed to compare the displacement between two subject groups in each axis. I understand the three coordinates as the same data, so you should use a two-way ANOVA. The same in the physical properties (Elastic and Viscosity parameters) because I understand that they are two parameters of the same Voigt model.

6. PLOS authors have the option to publish the peer review history of their article (what does this mean?). If published, this will include your full peer review and any attached files.

Reviewer #1: No

Reviewer #2: No

---

## [Author Response · Author response to Decision Letter 0]

23 Mar 2021

Dear Editor, 

Thank you for giving us the opportunity to improve and resubmit our manuscript “Functional impairment in facial expression generation in patients with repaired unilateral cleft lip: Effects of the physical properties of facial soft tissues”. Please find enclosed the revised manuscript for further consideration. The manuscript has been revised according to the comments raised by the reviewer to the best of our ability. Changes to the manuscript are highlighted in red. Please find our detailed replies to the reviewers’ comments attached with this revision. We would like to thank the reviewers for their constructive and competent criticism, and we hope that our revised manuscript will be acceptable for publication in PLOS ONE. 

Response to the academic editor and the reviewers:

Academic editor:

General comments:

Thank you for submitting your manuscript to PLOS ONE. After careful consideration, we feel that it has merit but does not fully meet PLOS ONE’s publication criteria as it currently stands. Therefore, we invite you to submit a revised version of the manuscript that addresses the points raised during the review process.

Comment 1)

Although the reviewers’ comments are short, they are critically important. The reviewers and I are not 100% convinced about the correctness of the statistical analysis. Please, address the reviewers’ comments on statistical analyses and, if necessary, expand the explanations to support their use. I would like to give you the opportunity to check that issue and explain yourself in the rebuttal letter; however, note that if the doubts about statistics persist after review, I may need consult with additional reviewers or with the PLOS ONE statistics support to make sure the analysis is sound.

Response 1)

Thank you for your thoughtful comment. We have answered Reviewer #2 and tried to any possible concerns that Reviewer #2 expressed. However, Reviewer #2 considered that a two-way ANOVA would be appropriate for this situation. If necessary, we can re-consider our position on this and are happy to seek support in relation to the statistical analysis if it is available.

Comment 2)

Reviewers also disagree on data availability, please check if you comply with PLOS ONE policy on data sharing.

Response 2)

Thank you for your comment. Accordingly, the original data is also published in a data repository service (doi:10.5061/dryad.h44j0zpjp) according to the data policy of PLOS ONE (https://datadryad.org/stash/share/4oR3jP3nzg869G6fxn7U0G36GolXhZr9dhCu6A87_r8. ).

Comment 3)

Finally, take into consideration the suggestions of Reviewer 2 on extending basic explanations in the main text that are now too succinct. Think that PLOS ONE is a generalist outlet with a broad audience.

Response 3)

Thank you for your comment. We have included additional information for your broad audience.

Reviewer #1: 

General Comments:

This paper were analyzed facial morphology while smiling in patient with unilateral cleft lip and palate. Correlation between facial displacement and physical property is interesting and should be beneficial information for readers of the journal. However, there is a room for improvement as follows;

Comment 1)

Title, Authors started the title with Functional impairment. However, the movement mentioned in this study was just for smiling and at rest position. The movement written in this study technically was not the function. Please reconsider about the title.

Response 1)

We thank the reviewer for this important suggestion. Accordingly, we have changed the title to “Impairment in facial expression generation in patients with repaired unilateral cleft lip: Effects of the physical properties of facial soft tissues”.

Comment 2)

P9L2; Authors recorded each expression once. Reproducibility of each position was not mentioned even if participants were trained several times. Reproducibility should be added.

Response 2) 

Thank you for your comment. We have added the intraclass correlation coefficients and 95% confidence intervals for the minimal detectable change of the two facial expressions. (Page 9)

Comment 3)

P14L1-2; Expression should be revise.

Response 3)

Thank you for your comments. We have deleted, “For details, please see our previous study [19, 22].” Instead, we added a brief explanation (Page 14).

Comment 4)

P40L6; Facial expression evaluated in this study was only for the maximum smiling. However, there should be more expressions other than maximum smiling. It would be too hasty to reach the clinical options including surgery from these results. Please reconsider or add the discussion about this point.

Response 4) 

Thank you for bringing this to our attention. Accordingly, we have added this as a limitation of the present study (Page 41).

Reviewer #2: 

General Comments:

This is a clinical experiment where they test that physical properties of the scar and surrounding facial soft tissue might affect facial displacement while smiling in patients with UCLP. I find the work interesting and the research well performed but I think that needs a better writing.

Comment 1)

I am familiar with clinical journals and I know that most of the works are written systematically, but PLoS ONe is not a clinical journal and, therefore, some of the text needs a better explanation. For example, you cannot write "For details, please see our previous study". Instead, you need to explain a summary of the method and cite the study. The manuscript itself should explain all the details of the research to make it repeatable for others. The methods part needs to be more explained. 

Response 1) 

Thank you for your comments. We have deleted, “For details, please see our previous study [19, 22].” Instead, we added a brief explanation (Page 14).

Comment 2)

I discovered that to obtain the Elastic and Viscosity parameters you were using a device because it is explained in the legend of the figure. This should be explained in the core text. Moreover, I went to other publication to guess that elastic and viscoelastic parameters were based in the Voigt equation. This should be explained briefly in the text, too. And cite the reference as well. Moreover, the method used to compute the Elastic and Viscosity parameters using the Vesmeter device is defined in one reference, but you should -at least- explain the basics in these documents. Especially because the information you can find in Internet provided by the manufacturer of this device is only in Japanese.

Response 2) 

Thank you for your comments. We have cited English documents explaining the instrument [11] in the Introduction. We also added a brief explanation about the instrument to the main text (Page 11).

Comment 3)

I have a second concern about the results and their statistical treatment. For example, I am not sure that a two-sample t-test can be performed to compare the displacement between two subject groups in each axis. I understand the three coordinates as the same data, so you should use a two-way ANOVA. The same in the physical properties (Elastic and Viscosity parameters) because I understand that they are two parameters of the same Voigt model.

Response 3) 

Thank you for the comments. To the best of our knowledge, a two-way ANOVA would be inappropriate in this situation because X-, Y-, Z-axis values were related values but has a completely different definition. To clarify this issue, we attached Tables A, B, and C (in a separate file) to exemplify data that would be appropriate for a two-way ANOVA data and our data. When compared the data to Table A, showing the data that be applied to a two-way ANOVA, the data in Tables B and C cannot be applied to this analysis because X, Y, Z-displacements cannot be aligned in one column, as each value has a different meaning. Furthermore, when considering weight, height, and chest circumference in the two groups, it is difficult to apply a two-way ANOVA to confirm the interaction between groups and variables. We consider that the weight, height, and chest circumference are similar values to the X-, Y-, Z-values in the present study. As such, we are afraid that the application of a two-way ANOVA would not be appropriate.

However, we assumed that the reviewer was concerned with two possibilities. The first is related to possible associations among the displacement variables in the X-, Y-, Z-axes; the second would be the family-wise type I error rates. Even though we could not confirm the reviewer’s intent, to eliminate any possible concerns, we included two additional statistics. For the latter, we controlled for the false discovery rate using the Benjamini-Hochberg procedure for statistics in facial displacements and facial physical characteristics (Elasticity and Viscosity). For the former, we performed a linear regression analysis to classify two groups when the X-, Y-, Z-values were used as dependent variables and the group was used as an independent variable for each semi-landmark (S2 Fig). This is for considering any associations among the three axis values. The significance of the regression analysis and the significant dependent variable is shown in S3 Fig; the results almost coincided with the previous results, with the exception of the transverse displacement of the ala. We hope that this meets the reviewer's expectations. If the reviewer has any further concerns regarding the details, we are prepared to perform additional analyses or comments.

 

Journal Requirements:

Comment 1:

Response 1:

Noted.

Comment 2:

Figures Include Patient Photos – Patient Consent for Publication

We note that Figures S1, S2 and S3 and Figures 1 & 2 include images of participants in the study.

As per the PLOS ONE policy (http://journals.plos.org/plosone/s/submission-guidelines#loc-human-subjects-research) on papers that include identifying, or potentially identifying, information, the individual(s) orparent(s)/guardian(s) must be informed of the terms of the PLOS open-access (CC-BY) license and provide specific permission for publication of these details under the terms of this license. Please download the Consent Form for Publication in a PLOS Journal (http://journals.plos.org/plosone/s/file?id=8ce6/plos-consent-form-english.pdf). The signed consent form should not be submitted with the manuscript, but should be securely filed in the individual's case notes. Please amend the methods section and ethics statement of the manuscript to explicitly state that the patient/participant has provided consent for publication: “The individual in this manuscript has given written informed consent (as outlined in PLOS consent form) to publish these case details”. If you are unable to obtain consent from the subject of the photographs, you will need to remove the figures and any other textual identifying information or case descriptions for these individuals.

Response 2:

All consent forms are attached now. We had deleted Figs S1 and S2 because we could not have new consent forms from these participants.

Comment 3:

You indicated that the inclusion criteria for participants within your study is 15–37 years old. In your Methods section, please ensure you have also stated whether you obtained consent from parents or guardians of the minors included in the study or whether the research ethics committee or IRB specifically waived the need for their consent.

Response 3: We have included that we obtained consent from parents or guardians of the participants under 20 years old included in the study

Comment 4:

In your Data Availability statement, you have not specified where the minimal data set underlying the results described in your manuscript can be found. PLOS defines a study's minimal data set as the underlying data used to reach the conclusions drawn in the manuscript and any additional data required to replicate the reported study findings in their entirety. All PLOS journals require that the minimal data set be made fully available. For more information about our data policy, please see http://journals.plos.org/plosone/s/data-availability. Upon re-submitting your revised manuscript, please upload your study’s minimal underlying data set as either Supporting Information files or to a stable, public repository and include the relevant URLs, DOIs, or accession numbers within your revised cover letter. For a list of acceptable repositories, please see http://journals.plos.org/plosone/s/data-availability#loc-recommended-repositories. Any potentially identifying patient information must be fully anonymized. Important: If there are ethical or legal restrictions to sharing your data publicly, please explain these restrictions in detail. Please see our guidelines for more information on what we consider unacceptable restrictions to publicly sharing data: http://journals.plos.org/plosone/s/data-availability#loc-unacceptable-data-access-restrictions. Note that it is not acceptable for the authors to be the sole named individuals responsible for ensuring data access. We will update your Data Availability statement to reflect the information you provide in your cover letter.

Response 4:

Noted.

---

## [Editor Report · Decision Letter 1]

29 Mar 2021

Impairment in facial expression generation in patients with repaired unilateral cleft lip: Effects of the physical properties of facial soft tissues

PONE-D-20-33973R1

Dear Dr. Tanikawa,

We’re pleased to inform you that your manuscript has been judged scientifically suitable for publication and will be formally accepted for publication once it meets all outstanding technical requirements.

Kind regards,

Borja Esteve-Altava, Ph.D.

Academic Editor

PLOS ONE

Additional Editor Comments (optional):

I am satisfied about how the authors responded and implemented the reviewers' suggestions. I only advice to double check that the repository with the raw data is available because I could not access it myself using the link provided. 
---

## [Editor Report · Acceptance letter]

5 Apr 2021

PONE-D-20-33973R1 

Impairment in facial expression generation in patients with repaired unilateral cleft lip: Effects of the physical properties of facial soft tissues 

Dear Dr. Tanikawa:

I'm pleased to inform you that your manuscript has been deemed suitable for publication in PLOS ONE. Congratulations! Your manuscript is now with our production department. 

Kind regards, 

on behalf of

Dr. Borja Esteve-Altava 

Academic Editor

PLOS ONE